# Mental Stress Assessment Using Ultra Short Term HRV Analysis Based on Non-Linear Method

**DOI:** 10.3390/bios12070465

**Published:** 2022-06-27

**Authors:** Seungjae Lee, Ho Bin Hwang, Seongryul Park, Sanghag Kim, Jung Hee Ha, Yoojin Jang, Sejin Hwang, Hoon-Ki Park, Jongshill Lee, In Young Kim

**Affiliations:** 1Department of Biomedical Engineering, Hanyang University, Seoul 04763, Korea; seungjaelee@hanyang.ac.kr (S.L.); hobin0215@hanyang.ac.kr (H.B.H.); seongryul@hanyang.ac.kr (S.P.); 2Department of Sociology, Hanyang University, Seoul 04763, Korea; sanghag@hanyang.ac.kr; 3Graduate School of Counseling Psychology, Hanyang University, Seoul 04763, Korea; hajung366@hanyang.ac.kr (J.H.H.); yoojinjang@hanyang.ac.kr (Y.J.); 4Department Anatomy and Cell Biology, College of Medicine, Hanyang University, Seoul 04763, Korea; hwangsj@hanyang.ac.kr; 5Department of Family Medicine, Hanyang University, Seoul 04763, Korea; hoonkp@hanyang.ac.kr

**Keywords:** heart rate variability (HRV), ultra-short-term HRV analysis, empirical mode decomposition (EMD), non-linear features, stress assessment

## Abstract

Mental stress is on the rise as one of the major health problems in modern society. It is important to detect and manage mental stress to prevent various diseases caused by stress and to maintain a healthy life. The purpose of this paper is to present new heart rate variability (HRV) features based on empirical mode decomposition and to detect acute mental stress through short-term HRV (5 min) and ultra-short-term HRV (under 5 min) analysis. HRV signals were acquired from 74 young police officers using acute stressors, including the Trier Social Stress Test and horror movie viewing, and a total of 26 features, including the proposed IMF energy features and general HRV features, were extracted. A support vector machine (SVM) classification model is used to classify the stress and non-stress states through leave-one-subject-out cross-validation. The classification accuracies of short-term HRV and ultra-short-term HRV analysis are 86.5% and 90.5%, respectively. In the results of ultra-short-term HRV analysis using various time lengths, we suggest the optimal duration to detect mental stress, which can be applied to wearable devices or healthcare systems.

## 1. Introduction

As the number of people with health problems due to stress increases every year, mental stress is emerging as one of the major human health problems in modern society. Mental stress not only harms mental and physical health, but also causes various diseases, such as diabetes, cardiovascular and respiratory diseases, depression, and cancers [1,2,3,4]. Mental stress is largely classified into chronic stress and acute stress. When one faces an acute stressor, the human body triggers a “fight-or-flight” response, a survival mechanism triggered by external stimuli to maintain homeostasis. This response activates the sympathetic nervous system within the body’s autonomic nervous system (ANS) and causes changes in the body through the endocrine system [5]. If an acute stressor continuously affects the body, it can be transformed into chronic stress, in which the ANS becomes unbalanced. Therefore, it is necessary to quantitatively measure and manage acute stress in daily life to remain healthy.

New methods for measuring stress are required in various fields. First, questionnaire methods [6,7,8] are often used in the medical field, but such methods can be considered a subjective indicator of each individual. Another stress measurement method utilizes biomarkers, such as salivary alpha-amylase and cortisol [9,10]. Although the biomarker method is an objective stress measurement method, it has disadvantages in that continuous stress monitoring is impossible, and measurement is inconvenient. Stress measurement using physiological signals can compensate for the shortcomings of other methods and has been actively studied in recent years to detect the physiological responses that change due to the presence of a stressor. Stress measurement studies are mainly conducted on physiological signals, such as ECG [11,12], GSR [13,14], and EEG [15,16], and some studies have described the combined use of several physiological signals [17,18].

Well-described methods of inducing acute stress in humans include the Stroop color-word test [19], the Trier Social Stressor Test (TSST) [20], mental arithmetic tasks [21], public speaking [22], Montreal Imaging Stress Task [23], and horror movie viewing [24]. According to a review of acute stressors [25], the most effective method for inducing stress among the various acute mental stressors is the TSST including public speaking and cognitive tasks, and there are TSST studies that conducted stress classification based on a single or multiple physiological signals. The research to classify stress and non-stressed states based on a single physiological signal is as follows. A study was conducted using HRV signals and human stress assessments, resulting in an accuracy of 84.38% [11]. In a study using EEG signals, features including absolute power, relative power, coherence, phase lag, and amplitude asymmetry were extracted for each frequency band using a 4-channel EEG wearable device. When non-stress and stress groups were classified, the highest reported accuracy was 98.76% [15]. Another stress classification study using GSR signals obtained an average classification accuracy of 94.62% based on the cvxEDA method, which is a rigorous and robust model [14]. Most previous studies were conducted using a single physiological signal, such as EEG or GSR; have a small number of subjects or data imbalance; and their application might not be suitable for real-life scenarios. In studies considering multiple physiological signals, stress classification results using EEG, GSR, and PPG achieved 79% accuracy; classification accuracy of 94% was achieved when the aforementioned measurements were fused with sociometric sensors [17]. Another study using EEG, GSR, and PPG achieved accuracies of 87.5% and 96.25% when EEG only and all three physiological signals were considered, respectively [18]. Although the performance of the previous stress classification studies has higher accuracy when using multiple signals rather than a single signal, it would be difficult to assess mental stress using multiple physiological signals in real-world situations.

HRV, which is used as a marker related to the ANS, is mainly used in studies of stress assessment. Much of the mental stress assessment research analyzes HRV signals recorded at 5 min intervals, defined as short-term HRV analysis [26]. As the demand for real-time stress monitoring increases with the development of wearable devices [27], recent healthcare systems require analysis of HRV signals shorter than 5 min, which is defined as ultra-short-term HRV analysis [26]. Recently, stress classification studies have been conducted using ultra-short-term HRV analysis [28,29,30]. These studies use HRV features related to the ANS. Since short-term and ultra-short-term HRV analyses have different lengths of data, and it is necessary to validate the ultra-short-term HRV features as short-term features. Among stress-related HRV features, frequency domain features are greatly affected by data length. In order to measure high-frequency (HF) and low-frequency (LF) spectrum power, HRV data is required for a duration of at least 60 and 250 s, respectively [26]. Moreover, approximate entropy, one of the non-linear methods, cannot be used for data representing less than 3 min [31]. Castaldo et al. [32] suggested that ultra-short-term HRV features are surrogates of short-term features at different time lengths during mental stress. Since the HRV signal is non-linear and non-stationary due to the dynamics of the complex cardiac system, it is appropriate to use Empirical Mode Decomposition (EMD), a time-frequency domain signal method with non-stationary and non-linear characteristics. There are studies that have conducted stress analysis of HRV signals using the EMD method [33,34,35]. There are also studies that analyzed HRV signal using entropy methods with non-linear properties, including approximate entropy [36], sample entropy [37], and permutation entropy [38].

From this point of view, we analyzed intrinsic mode function (IMF) energy features extracted from EMDs that can provide frequency domain information using less than 5 min of data. Furthermore, we evaluated the stress classification accuracy of short-term and ultra-short-term HRV data using time domain features, including general HRV features, entropy features, and proposed EMD-based features, and suggested the optimal time length for ultra-short-term HRV data collection in acutely stressful situations. In addition, the classification method used a linear SVM classifier, and the appropriate evaluation was performed with the leave-one-subject-out cross-validation (LOSOCV) method, which is most similar to real-life application.

## 2. Methods

The stress classification method proposed in this study is shown in Figure 1. Subject selection and the experimental protocol for HRV signal measurement are described in Section 2.1 and Section 2.2, respectively; preprocessing to remove ectopic heartbeats is described in Section 2.3; the description of the proposed HRV features and general features is outlined in Section 2.4; and ranked feature processing to increase accuracy and a classification method using leave-one-subject-out cross-validation are outlined in Section 2.5.

### 2.1. Dataset

The stress database [39,40] provided by PhysioNet assumes that the subject is stressful in the given stress situations, and this approach may be limited in classifying stress. In addition, the PhysioNet stress database is difficult to generalize because it consists of a small number of subjects. In order to supplement the above limitations, the following experiment was conducted.

#### 2.1.1. Subjects

The dataset consists of physiological signals and questionnaires from 80 participants (78 males and 2 females). All participants were third-year police officers without heart disease and were in good physical condition. The data for six subjects (all male) were excluded due to sensor problems. This study was approved by the Institutional Review Board of Hanyang University Hospital and was conducted according to the guidelines of the Declaration of Helsinki, and all subjects provided informed consent before the experiment (HYUIRB-202009-032-3)

#### 2.1.2. Experimental Protocol

All participants performed the stress-inducing experiment in a laboratory environment according to the experimental protocol shown in Figure 2. The sequence of the experiment protocol was sensor attachment, resting state (pre) for 5 min, exposure to each of the 3 stressors for 5 min, resting state (post) for 5 min, and self-reported subjective stress intensity. We used various methods to induce acute mental stress, including TSST (public speaking and arithmetic task) and horror movie viewing.

Before the experiment, participants wore a Polar H10 HR monitor with a Polar Pro Chest Strap (Polar Electro Oy, Kempele, Finland) to measure respective heart rate (HR) and heart rate variability (HRV), which represent autonomic responses. The RR interval data were transferred to Samsung Galaxy Tab A (Samsung Electronics, Co., Ltd., Seoul, South Korea) and saved in Polar’s app. Based on findings that alpha-amylase increases when humans are exposed to a stressful situation [41,42], salivary alpha-amylase was also measured using a COCORO Meter (Nipro Co, Osaka, Japan).

As shown in Figure 2, the stress-inducing experiment protocol consists of a non-stress section, which is comprised of rest (pre and post), and a stress section, comprised of TSST (public speaking, arithmetic task) and horror movie viewing. In the resting section, the participants were seated on a comfortable chair, and the HRV signal was measured for 5 min of relaxation. During the Stress 1 section, public speaking including job-related questions and a simulated job interview that was conducted for 5 min. In the Stress 2 section, mental arithmetic tasks (MATs) were performed. The MAT was to continue subtracting 17 from 2023. If an answer was wrong, subtraction started again from 2023, and the MAT was repeated for 5 min. During the horror movie viewing within the Stress 3 section, participants watched horror movie clips for 5 min. After the experiment, a questionnaire was conducted on the ranking of subjective stress intensity. The ranking of subjective stress intensity was set to 5 for the most stressful situations and 1 for the least stressful situations. 59 of 74 subjects answered that mental arithmetic was the most stressful, while the remaining eight and seven subjects said horror movie clips and public speaking were the most stressful, respectively.

### 2.2. Data Preprocessing

#### 2.2.1. Noise Removal and Interpolation of HRV Signal

Prior to HRV analysis, preprocessing is essential to remove the outliers in RR intervals caused by noise, such as movement. Outliers in RR interval data were removed and defined as data outside 3 standard deviations (SD) from the mean [43]. Considering the non-linear characteristics of HRV signals, cubic spline interpolation was performed [44]. The HRV signal used to obtain EMD-based features was interpolated with cubic spline and was resampled at 8 Hz.

#### 2.2.2. Comparison of Short-Term HRV and Ultra-Short-Term HRV

For comparison with ultra-short-term HRV, we analyzed short-term HRV using data collected over 5 min, which is the data length commonly used in previous studies. We selected the resting state and stress state for each participant using the rankings of subjective stress intensity. The resting state was selected as the lowest subjective stress intensity ranking between Resting 1, measured before the stress experiment, and Resting 2, measured after the stress experiment. The stress state was selected as the highest stress ranking among Stress 1, Stress 2, and Stress 3.

According to Thomas Wyss et al. [45], the HR, an ANS indicator, is initially high during acute mental stress situations and decreases as the stress situation continues. These results indicate that the response of the sympathetic nervous system to acute mental stress not only responds rapidly, but also adapts to a stress stimulus due to ANS homeostasis. As shown in Figure 3, ultra-short-term HRV analysis was performed by dividing time segments into first, middle, and last segments over 1, 2, and 3 min, respectively. Each resting state was paired with stress state, and the segment with the lowest average HR was used.

### 2.3. Feature Extraction

#### 2.3.1. Time Domain Features

Time domain HRV features frequently used to evaluate acute mental stress were extracted. The mean RR interval (RR), standard deviation of RR interval (SDNN), square root of the mean squared difference between successive RR intervals (RMSSD), and the proportion of successive differences between RR intervals greater than x *msec* (pNN30, pNN50) were extracted. Expressions for these features are as follows.
(1)$$\mathrm{SDNN}=\sqrt{\frac{1}{N-1}(\overset{N-1}{\underset{i=1}{\sum }}{\left(\mathrm{RR}i-mean\mathrm{RR}\right)}^{2})}$$
(2)$$\mathrm{RMSSD}=\sqrt{\frac{1}{N-1}\overset{N}{\underset{i=1}{\sum }}{\left({\mathrm{RR}}_{i+1}-{\mathrm{RR}}_{i}\right)}^{2}}$$
(3)$$\mathrm{pNNx}=\frac{\;NNx}{N-1}*100$$

Additionally, we extracted G-pNNx (Grouped-pNNx), a new feature that can be applied to a group of young and healthy people. Based on a prior study [46], using a parameter between pNN10-40 instead of pNN50 is more suitable for stress assessment. We developed G-pNNx as a pNNx-based stress feature optimized for young people using data collected from young police officers at rest. The procedure for calculating the x-value of G-pNNx is as follows.

On the basis of subjectively ranked self-reported stress intensity in the experimental protocol, select the lowest stress rank between Resting 1 and Resting 2.If the distribution of data satisfies normality, obtain G-pNNx using the mean value of the distribution; otherwise, obtain it using the median value.

Since the data do not satisfy normality (Kolmogorov–Smirnov, *p* < 0.05) in our study, the x value of G-pNNx was obtained using the median value (18.5).

#### 2.3.2. EMD-Based Features

##### EMD

EMD is an adaptive time-series analysis method suitable for processing non-stationary and non-linear series [47]. Since the HRV signal has non-stationary and non-linear characteristics [48], EMD is suitable for our research. EMD can decompose any signal with an IMF. As the x-value in an IMF increases, the low-frequency component of the original signal is included. For example, IMF1 represents the highest local frequency component of the signal. In order to extract IMFs from the original signal using the EMD method, the following two basic conditions are essential [49].

1. The numbers of extrema and zero crossings must be the same or different at most by one within the entire dataset.

2. The mean value of the envelope defined by local maxima and minima must be zero at any point.

If the two conditions are satisfied, the IMF can be continuously decomposed by the EMD method for *x(t)* and mathematically expressed as follows.
(4)$$x\left(t\right)=\overset{n}{\underset{k=1}{\sum }}{\mathrm{IMF}}_{k}+r_{n}\;$$
where *x(t)* is the HRV signal, we used decomposed IMF of 1~3, and the residual *r* was not used.

##### Entropy Features

The entropy method is suitable for HRV signals as a non-linear method similar to EMD, and the entropy methods used in our study are permutation entropy and sample entropy.

Permutation entropy (PE) is useful to represent the complexity of dynamic time-series signals and has the advantages of simple calculation and robustness to noise [50]. PE is calculated by the following equation [51].
(5)$$PEn=-\overset{m!}{\underset{i=1}{\sum }}A_{i}log\left(A_{i}\right)\;$$

The factorial calculated from sequence length *m* (dimension) is the number of possible permutation patterns, and *A_i_* is the probability of the *i*-th permutation pattern. The setting values in the permutation entropy method are time delay *tau* and dimension *m*. Since the sampling frequency of the resampled HRV signal is 8 Hz, the maximum value of *tau* was set to 4 by the Nyquist–Shannon sampling theorem [52]. Since the minimum length of data used for ultra-short-term HRV analysis was *N* (data length) = 480 (sampling frequency = 8 Hz, 60 s), *m* was set to 4, suitable for values under the condition of *N* > 5*m*! [53].

Sample entropy compensates for the disadvantage of approximate entropy [54] and measures the irregularity and complexity of HRV signal and IMF components. The setting values used in sample entropy consist of embedding dimension *m* and tolerance *r*. In a previous study [55], the embedding dimension value was set to 2, and the tolerance value was set using the standard deviation of the HRV data (*r* = 0.2 × SD).

##### Energy Features

Previous stress assessment studies based on HRV signals frequently use frequency domain features, including the VLF component (~0.04 Hz), LF component (0.04~0.15 Hz), HF component (0.15~0.4 Hz), and LF/HF ratio. A major drawback of frequency domain features is the inaccuracy when using ultra-short-term HRV signals. According to [56], LF band (0.04~0.15 Hz) power requires HRV data to span at least 250 s from a theoretical point of view, so ultra-short-term HRV analysis using HRV data lasting about 4 min or less has poor reliability regarding several HRV features based on LF band power.

Based on Parseval’s theorem [57] that the total energy of a signal can be calculated by summing power across time or spectral power across frequency, as shown in Equation (6), this paper presents a methodology that can replace frequency domain features with the energy of IMF components. The methodology compares the HF band power and energy of the high-frequency component IMF1 and the LF band power and energy of the relatively low-frequency components (IMF2 and IMF3). In addition, IMF energy features corresponding to the LF/HF ratio, normalized-LF, and normalized-HF were extracted.
(6)$$\overset{\infty }{\underset{n=-\infty }{\sum }}{\left|x\left[n\right]\right|}^{2}=\frac{1}{2\pi }\int_{-\pi }^{\pi }{\left|X_{2\pi }\left(\phi \right)\right|}^{2}d\phi$$

##### SD-IMF and RMSSD-IMF Features

Three IMF components obtained through the EMD method were extracted using SDNN Equations (1) and RMSSD (2). Instead of RRi in Equation (1), three SD-IMF features were used for the IMF components, and three RMSSD-IMF features were extracted using Equation (2).

### 2.4. Feature Ranking Method

The number of extracted HRV features was 26, including 6 general time domain features, 3 entropy/energy features of HRV signal not decomposed through EMD (permutation entropy, sample entropy, and energy), 6 EMD-based entropy features, 5 EMD-based entropy features, and 6 EMD-based time domain features.

The Relief-F algorithm was applied to remove features that did not contribute to stress classification performance and to improve computational efficiency. The Relief-F algorithm is a method to evaluate the contribution of each feature based on k-nearest neighbors by increasing the inter-class difference and decreasing the intra-class difference [58]. The algorithm finds a sample with the same class label and a sample with a different class label closest to a randomly selected sample among the k samples in the closest order and calculates a weight using the difference between the selected sample and the closest sample. In this study, the feature with the lowest weight value between the resting state and the stress state was removed sequentially to achieve good performance.

### 2.5. Classification Method

#### 2.5.1. Support Vector Machine (SVM) Classifier

In order to classify the resting and stress states, the SVM classifier widely used in stress research was used. The SVM classifier is a method of finding the optimal hyperplane to classify a class, and a vector contributing to creating the hyperplane is called a support vector [59]. Mathematically, an SVM is represented as follows [60]:(7)$$H\left(x\right)=sign\left[\overset{p}{\underset{k=1}{\sum }}\alpha_{k}t_{k}P\left(z,z_{k}\right)+b\right]\;$$
where *P*(*z*,*z_k_*) is the kernel function, *z_k_* is the D-dimension k-input vector (feature), *t_k_* is the target class vector, *a_k_* is the LaGrangian multiplier, and *b* represents the bias term. We used SVM with a linear classifier and classified stress and non-stress states using short-term HRV and ultra-short-term HRV data.

#### 2.5.2. Leave-One-Subject-Out Cross Validation (LOSOCV)

The LOSOCV method is a variation of the k-fold cross-validation approach that validates as many folds as there are subjects included in the data set [61]. LOSOCV evaluates the accuracy on new subjects that have not been seen by the model, suggesting whether the developed classifier model can achieve classification performance when applied in real situations. In this study, the performance of the training model was evaluated using LOSOCV.

#### 2.5.3. Performance Evaluation

We calculated evaluation indicators, such as accuracy, precision, recall, and F1-score, to explain the results of LOSOCV. Each evaluation indicator is expressed as follows [62]:(8)$$Accuracy=\frac{TP+TN}{TP+TN+FP+FN}\;$$
(9)$$Precision=\frac{TP}{TP+FP}$$
(10)$$Recall=\frac{TP}{TP+FN}\;$$
(11)$$F1\;Score=2*\frac{Precision*Recall}{Precision+Recall}\;$$

## 3. Results

### 3.1. Relationships between Frequency Domain Features and IMF Energy Features

In order to perceive the trends in 5-min HRV signal and IMF components in the frequency domain, we represented each component using fast Fourier transforms (FFTs), as shown in Figure 4. The HF band (0.15–0.4 Hz) of the HRV and IMF1 spectral areas (blue) are similar, and the LF band (0.04–0.15 Hz) of the HRV spectrum and the sum of the IMF2 (red) and IMF3 spectral areas (pink) are similar. In addition, the mean frequency for each IMF using the entire data including the resting and stress state is shown in Figure 5. The mean frequency of IMF1 is included in the HF band, and IMF2 and IMF3 are included in the LF band.

The Pearson correlation method was used to describe the proposed IMF energy features as surrogates of frequency domain features. Table 1 presents the correlation coefficients of the relationships between IMF energy (IMF1, IMF2+IMF3, and ratio) and frequency domain features (HF, LF, and HF/LF ratio). Using the scale of Hopkins [63], the results in Table 1 were qualitatively analyzed with the Pearson correlation *r*, described as trivial (0.0–0.1), small (0.1–0.3), moderate (0.3–0.5), large (0.5–0.7), very large (0.7–0.9), or nearly perfect (0.9–1.0).

Using all data, including both the resting state and the stress state, IMF1-energy features had a nearly perfect correlation value (*r* = 0.93) with HF and a relatively low correlation value with LF (*r* = 0.77). Contrary to IMF1-energy features, the IMF2+IMF3-energy features had a nearly perfect correlation value (*r* = 0.92) with LF and a relatively low and very large correlation value (*r* = 0.79) in HF. The IMF-energy ratio was highly correlated with the LF/HF ratio (*r* = 0.86).

### 3.2. Comparison of Feature Ranks between Resting and Stress States

When classifying the resting state and stress state with the Relief-F algorithm using short-term HRV data, the results are listed in descending order of weight, as shown in Table 2. Linear SVM classifier training was repeated by adding features in ascending order from the ranked features one at a time, and the optimal number of features with the best classification performance was selected. From these results, the proposed energy-related features, such as ranks 1, 3, and 6, have high weights.

### 3.3. Short-Term HRV Classification and Performance Evaluation

The results of LOSOCV performance evaluation on short-term HRV data after feature selection are shown in Figure 6. The best classification performance was obtained when the classifier model was developed using the top 17 highest ranked features. The results comparing the stress state to the resting state were 86.5% accurate, with a recall of 85.1%, precision of 87.5%, and F1-score of 86.3%.

### 3.4. Ultra-Short-Term Classification and Performance Evaluation

The classification results based on ultra-short-term HRV data by dividing the time segments into each time length, designated as the first, middle, and last time lengths (1, 2, and 3 min), are shown in Figure 3. The linear-SVM classifier model was used in the same way as the classification method based on the short-term HRV data, and LOSOCV performance evaluation was performed using the ranked features listed in Table 2. The results of ultra-short-term HRV classification were compared for time segment and time length (Table 3). When compared in the time segments between the first, middle, and last time lengths, the classification accuracy of the first length is higher than that of the middle and last lengths, as shown in Figure 7. The accuracy of the first segment of 2-min and 3-min lengths is higher than the classification accuracy using short-term HRV data. Segments of 1-min length achieved the highest accuracy in the middle part, and this result indicated a different tendency than those of segments of 2-min and 3-min length. All segments of 1-min length led to less accurate predictions than the classification performance using short-term HRV data.

### 3.5. Salivary Cortisol of Different States

According to the experiment protocol in Figure 2, sAA measurements were performed a total of four times during the experiment (sAA1: After public speaking sAA, sAA2: After mental arithmetic sAA, sAA3: After horror movie sAA, sAA4: After resting sAA). Since not all sAA data satisfied the normality test, the resting state (sAA4) and stress states (sAA1~3) analyses were performed using the non-parametric Mann–Whitney U test. Compared with sAA4, sAA1 (*p* = 0.056), sAA2 (*p* = 0.797), and sAA3 (*p* = 0.082) were not significant in all stress states.

## 4. Discussion

This study suggests that the proposed IMF energy features obtained using EMD are good surrogates of the frequency domain features. Although there is previous research that showed similarity among HRV features, including time domain, frequency domain, and non-linear features [64], this is the first study to suggest that frequency domain features such as HF, LF, and ratio can be replaced with other HRV features.

Ultra-short-term HRV analysis, which evaluates mental stress using data less than 5 min long, is a topic of increasing interest [56]. Castaldo et al. [32] presented time segments and HRV features in ultra-short-term HRV data that can be replaced with short-term HRV analysis as a result of correlation analysis between ultra-short-term and short-term HRV features. It was suggested that features including HR, RMSSD, pNN50, and sample entropy can be substituted for ultra-short-term HRV analysis with a 2-min length compared to short-term HRV analysis (*r* > 0.7). Using the same method from the research above, 4 time lengths (5 min, 3 min, 2 min, and 1 min) were compared, as in Table 4. In the resting state, all features were significantly correlated in each time scale (*r* > 0.7), and all features except the 1 min time scale were significantly correlated in the stress state (*r* > 0.7). Accordingly, our proposed IMF energy features show that ultra-short-term HRV analysis can replace short-term HRV analysis with a time length of at least 2 min.

The proposed stress classification method was compared with some of the latest studies that selected TSST including public speaking and cognitive tasks as mental stressors and conducted acute stress classification with short-term HRV analysis. Table 5 compares the number of subjects, type of measured physiological signals, classifier model, validation method, and accuracy of our and previous studies. Multi-physiological signals, including ECG, PPG, and GSR, were measured using public speaking as a stressor, and the result was achieved with 79% accuracy using the AdaBoost classifier and four-fold cross-validation to classify between stress and non-stress states [17]. The results of this study have relatively low accuracy compared to other studies. The stress classification accuracy of another study [18] was 96.3% using leave-one-out cross-validation with the SVM-RBF classifier considering PPG, GSR, and EEG. However, the result considering a single signal (PPG) was 80.0%, which is lower than our accuracy of 86.5%. Another study [12] selected TSST and Stroop color test as stressors and used a random forest classifier and three-fold cross-validation. The results for overlapping and not overlapping HRV signals over 5 min are accuracies of 96.0% and 85.9%, respectively. The disadvantage of the aforementioned study is that the number of subjects is small, and leave-one-subject-out cross-validation, similar to that applied in real-world, was not used. The study most similar to the present study [11] selected TSST as the stressor, and the classification accuracy results of five-fold cross-validation and leave-one-subject-out cross-validation with an ANN classifier were 91% and 84.4%, respectively. When comparing based on LOSOCV, the results of our study achieved a higher accuracy of 86.5%.

The dataset of our study, collected from 74 people, is the largest study that selected TSST as a stressor. Compared with precedent studies using a single physiological signal and LOSOCV, the result of the proposed stress detection method presents the highest performance. For various stress-inducing tasks (public speaking, arithmetic, horror movie viewing), the stress state and resting state were selected as individual stress intensities through a questionnaire to rank subjective stress intensity. The reason for dividing the stress and resting states in this way is that the stress intensity for each task differs by person, which is also one of the advantages compared to previous studies.

As HRV analysis of data spanning less than 5 min becomes important in several healthcare applications, ultra-short-term HRV analysis has been actively studied in previous studies of HRV stress assessment using various methods included in mental arithmetic tasks [28], examination stress [11,32], and the Stroop color-word task [19,65]. However, in all previous studies, the HRV data used for ultra-short-term HRV analysis was divided according to the required duration or the central position of the 5 min HRV data. The proposed study is the first considering stress adaptation due to ANS homeostasis during stress-induced tasks. By comparing the results of ultra-short-term HRV stress classification consisting of three time segments (first, middle, and last) and three durations (1 min, 2 min, and 3 min) with general short-term HRV stress classification, stress adaptation was confirmed by deriving higher accuracy in the first segment than in the last segment. In addition, the accuracy of ultra-short-term HRV stress classification (Table 3) shows higher accuracy in the first segments of 2-min and 3-min lengths compared to the accuracy of short-term HRV stress classification (86.5%). The optimal time length to be used for stress assessment using ultra-short-term HRV data is 3 min, but a minimum length of 2 min is suggested.

## 5. Conclusions

In our study, we presented IMF energy features based on EMD as a surrogate of general frequency domain features in HRV analysis and suggested the optimal duration for ultra-short-term HRV analysis as a result of stress classification accuracy comparison between short-term and ultra-short-term HRV analysis. Performance evaluation was conducted using LOSOCV, most similar to real-world situations. The proposed study has high accuracy compared to previous studies using similar classification methods based on a single physiological signal.

This study has three contributions. First, we provide frequency domain information with data of less than 5 min. Through EMD-based features, it is possible to replace information about frequency domain that are difficult to use in ultra-short-term. Second, we identified stress adaptability over time. In this way, the ultra-short-term analysis would be more appropriate when evaluating stress than the commonly used short-term analysis. Finally, we propose an optimal time length for ultra-short-term HRV data collection in acutely stressful situations. It is possible to provide a comfortable stress analysis service for several healthcare applications through the proposed length.

## Figures and Tables

**Figure 1 biosensors-12-00465-f001:**
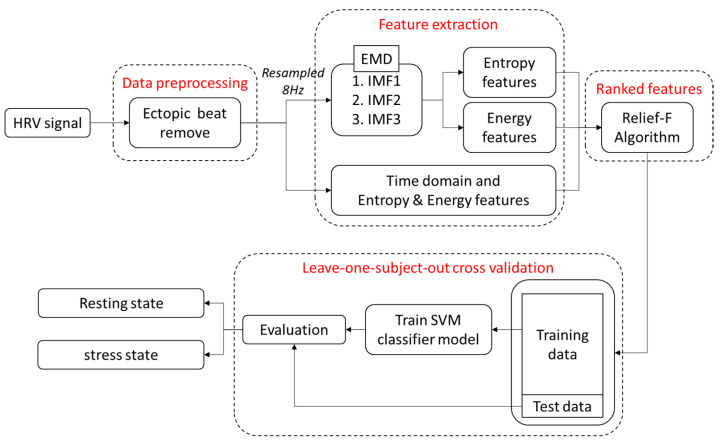
Flowchart for the proposed stress classification method.

**Figure 2 biosensors-12-00465-f002:**
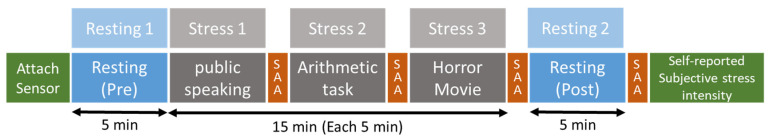
Experimental protocol for inducing stress.

**Figure 3 biosensors-12-00465-f003:**
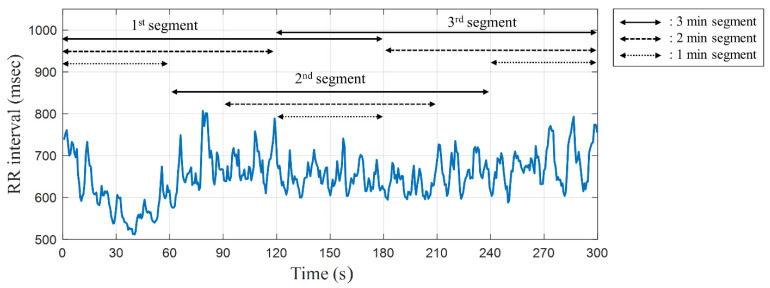
Ultra-short-term HRV explanation for each time segment during the experimental protocol.

**Figure 4 biosensors-12-00465-f004:**
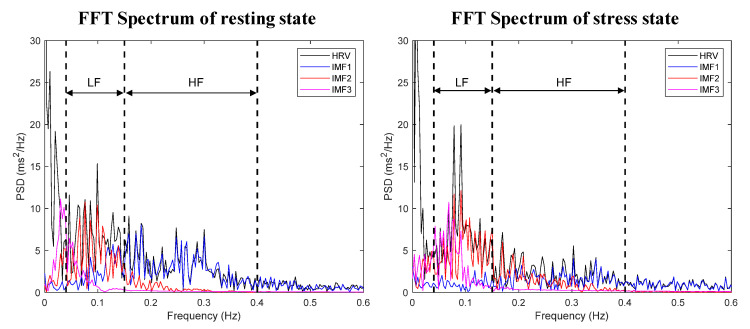
FFT spectra of resampled HRV and IMF components: resting state (**left**), stress state (**right**).

**Figure 5 biosensors-12-00465-f005:**
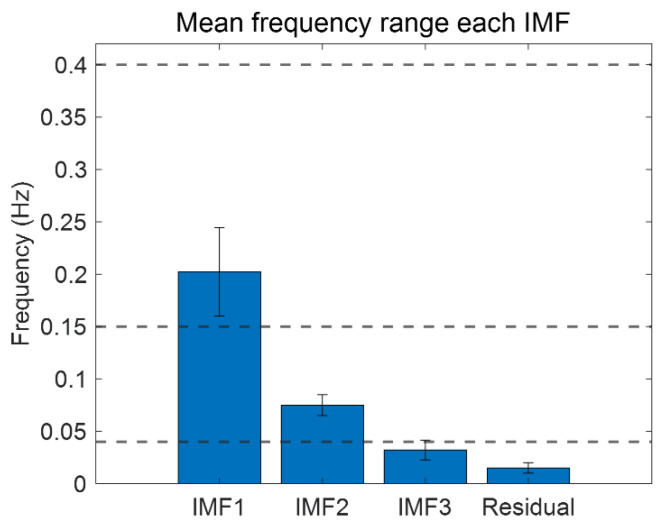
Mean frequency range of each IMF components in short-term HRV.

**Figure 6 biosensors-12-00465-f006:**
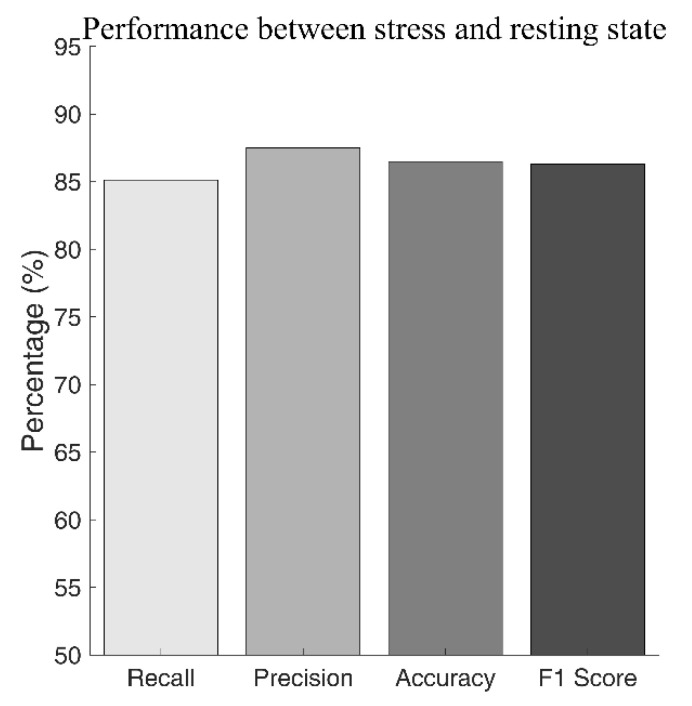
Classification performance using short-term HRV data.

**Figure 7 biosensors-12-00465-f007:**
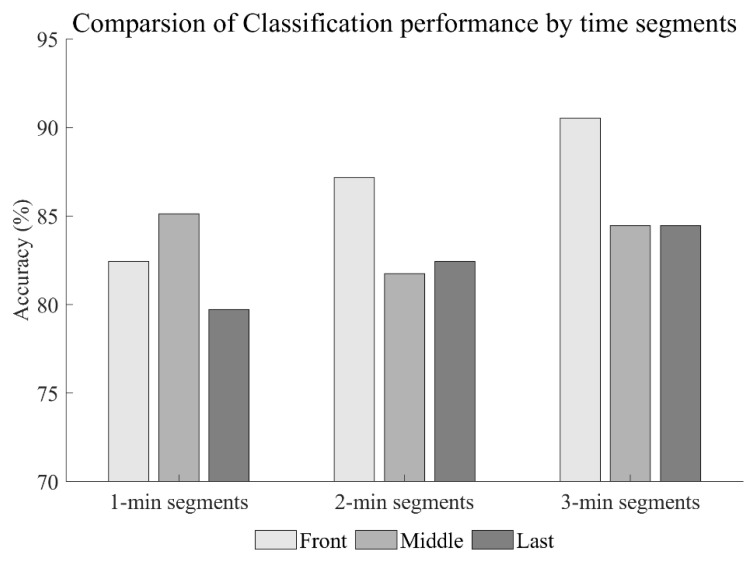
Comparison of classification accuracy using ultra-short-term HRV data.

**Table 1 biosensors-12-00465-t001:** Correlation coefficients between frequency domain features.

	HF	LF	LF/HF Ratio
$Energy_{IMF1}$	0.93	0.79	−0.29
$Energy_{IMF2+IMF3}$	0.77	0.92	−0.03
$\frac{Energy_{IMF2+IMF3}}{Energy_{IMF1}}$	−0.43	−0.09	0.86

**Table 2 biosensors-12-00465-t002:** HRV features in descending order of significance according to the Relief-F algorithm.

Rank	Feature Name	Rank	Feature Name
1	Energy	14	SpEn
2	SDNN	15	G-pNNx
3	Energy_(IMF2+IMF3)	16	PmEn_IMF3
4	SpEn_IMF3	17	pNN50
5	RMSSD_IMF3	18	pNN30
6	Energy_IMF1	19	SpEn_IMF2
7	SDNN_IMF3	20	Energy_IMF23/IMF1
8	HR	21	Energy_IMF1/IMF123
9	SDNN_IMF2	22	Energy_IMF23/IMF123
10	RMSSD_IMF2	23	PmEn
11	RMSSD	24	PmEn_IMF1
12	RMSSD_IMF1	25	PmEn_IMF2
13	SDNN_IMF1	26	SpEn_IMF1

**Table 3 biosensors-12-00465-t003:** Classification performance using ultra-short-term HRV data according to time segments and time lengths.

	Classification Performance (%)			
		Front	Middle	Last
3-min segments	Accuracy	90.5	84.5	84.5
	F1 Score	90.3	83.7	84.6
2-min segments	Accuracy	87.2	81.8	82.4
	F1 Score	86.7	82.1	81.9
1-min segments	Accuracy	82.4	85.1	79.7
	F1 Score	82.4	84.5	79.2

**Table 4 biosensors-12-00465-t004:** Correlation analysis of ultra-short-term vs. short-term HRV features.

	Rest State			Stress State		
HRV Features	3 vs. 5 min	2 vs. 5 min	1 vs. 5 min	3 vs. 5 min	2 vs. 5 min	1 vs. 5 min
${\mathrm{Energy}}_{\mathrm{IMF}1}$	**0.978**	**0.941**	**0.901**	**0.959**	**0.948**	**0.920**
${\mathrm{Energy}}_{\mathrm{IMF}2+\mathrm{IMF}3}$	**0.935**	**0.881**	**0.813**	**0.943**	**0.889**	**0.836**
$\frac{{\mathrm{Energy}}_{\mathrm{IMF}1}}{{\mathrm{Energy}}_{\mathrm{IMF}2+\mathrm{IMF}3}}$	**0.986**	**0.968**	**0.883**	**0.809**	**0.742**	0.674

**Table 5 biosensors-12-00465-t005:** Performance comparison of the proposed method with the state-of-the-art methods for short-term HRV analysis.

Paper	Number of Subjects	Physiological Signals (Modalities)	Classifier	Validation	Accuracy (Classes)
[17], 2017	18	ECG (HRV), PPG, GSR	AdaBoost	4-fold	79.0% (2)
[18], 2021	40	PPG PPG + GSR PPG + GSR + EEG	SVM-RBF	LOSOCV	80.0% (2) 86.3% (2) 96.3% (2)
[12], 2021	12	ECG (HRV)	Random Forest	3-fold	Non-overlapping: 85.9% (2) overlapping: 96.0% (2)
[11], 2020	57	ECG (HRV)	ANN	5-fold LOSOCV	91.0% (2) 84.4% (2)
**Proposed**	**74**	**ECG (HRV)**	**Linear SVM**	**LOSOCV**	**86.5% (2)**

## Data Availability

Data sharing is not applicable to this article.

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
