# Peer review of "Mental Stress Assessment Using Ultra Short Term HRV Analysis Based on Non-Linear Method"

_biosensors, 2022, doi:10.3390/bios12070465_

Round 1

Reviewer 1 Report

The article describes a mental stress assessment method by using ultra short term HRV analysis and based on support vector machine classifier.

Although the article is well written, easy to track and understand, its scientific contributions in terms of quantitative and qualitative aspects are difficult to find in the article.

 A precise measure of mental stress can be determined only by using a variety of instruments that have been designed to help measure individual stress levels. For this reason, researches interested in study the effect of acute stress response in humans require a valid and reliable acute stressor to be used under experimental conditions. In my opinion, from medical point of view, it is not enough to use the experimental protocol for inducing stress described in fig 2, by combining stress and a non-stress sections, and to use it with third-year police officers (trained to work in stressed conditions), without heart disease and in good physical condition.  

On the other hand, the HRV based analysis of ECG acquired signals from stressed persons are already studied deeply during the last years. The developed method described in the article describes only the use of these methods in some particular conditions, without any scientific contribution added to improve their accuracy. The comparison with other similar methods, available on scientific literature and described in the Table 5, cannot be considered because it is done by using different subjects with different stress types and levels. This is in my opinion, the main lack of the article. 

A more rigorous evaluation of the developed method can be obtained if it is applied on several internet available stress databases. 

Author Response

We appreciate the reviewers’ comments and editor’s suggestions. Our responses are given below, and the main text was modified based on the valuable comments from the referees.

Reviewer 2 Report

This paper presents a research article intitled ” Mental stress assessment using ultra short term HRV analysis based on non-linear method”. The paper proposed an approach to assess mental stress using ultra short term HRV analysis. Overall, the manuscript is well written and well organised. 

The introduction is very complete, providing sufficient background and include relevant references. The method is adequately described and the experimental design is appropriate.

The results and discussion are presented clearly but I suggest the improvement of conclusion.

Author Response

(The authors gave the same response as above.)

Reviewer 3 Report

This manuscript presented new heart rate variability (HRV) features based on empirical pattern decomposition. The results showed that the classification accuracy of short-term HRV and ultrashort-term HRV analysis was 86.5% and 90.5%, respectively. This work is interesting, and the authors provided sufficient experimental data and good results. So, this paper could be considered for publication in this journal. There are some questions that could be discussed.

(1)    What are the parameters that affect the function and properties of this sensing strategy?

(2)    What is the non-linear equation and detection limit?

(3)    The format of figures and tables should be uniform.

Author Response

(The authors gave the same response as above.)

Round 2

Reviewer 1 Report

Due to the positive answers given by the authors to the raised issues, now the article has an increased scientific content and it is ready, after solving the text formatting issues, to be included in the Journal.